# Preparation of a Novel Cellulose–Styrene Copolymer Adsorbent and Its Adsorption of Nitrobenzene from Aqueous Solutions

**DOI:** 10.3390/polym13040609

**Published:** 2021-02-18

**Authors:** Guifang Yang, Na Lin, Yuan Li, Xiaoxia Ye, Yifan Liu, Yuancai Lv, Chunxiang Lin, Minghua Liu

**Affiliations:** 1College of Chemical Engineering, Fuzhou University, Fuzhou 350116, China; M150410005@fzu.edu.cn (G.Y.); N190627042@fzu.edu.cn (N.L.); N180627037@fzu.edu.cn (Y.L.); yexiaoxia@fzu.edu.cn (X.Y.); yfanym@fzu.edu.cn (Y.L.); yclv@fzu.edu.cn (Y.L.); lcx2010@fzu.edu.cn (C.L.); 2Fujian Provincial Engineering Research Center of Rural Waste Recycling Technology, College of Environment & Resources, Fuzhou University, Fuzhou 350116, China

**Keywords:** cellulose, styrene, hydrophobicity, nitrobenzene, polymerization

## Abstract

A novel cellulose–styrene copolymer adsorbent (cellulose-St) was prepared using free radical polymerization. Successful polymerization was confirmed through Fourier Transform Infrared Spectroscopy (FTIR), Carbon 13 Solid Nuclear Magnetic Resonance (^13^C NMR) Spectroscopy, Scanning Electron Microscopy (SEM), etc. Cellulose-St possessed good hydrophobicity, and the best water contact angle of cellulose-St samples could reach 146°. It had the ability of adsorption for nitrobenzene (NB), and the adsorption process could be well described by the pseudo-second-order (R^2^ > 0.99) and three-stage intraparticle diffusion (R^2^ > 0.99) kinetic models. Furthermore, the dynamic adsorption experiments revealed that cellulose-St had the potential for continuous separation of NB in water, and the breakthrough point for the initial NB concentration of 10 mg/L reached 1.275 L/g. Moreover, cellulose-St exhibited excellent environmental adaptability that it could maintain its hydrophobicity and adsorption ability for NB in strong acids, strong alkalis, or organic solvents. The used cellulose-St could be reused after washing with ethanol and keep almost constant adsorption capacity after ten cycles.

## 1. Introduction

Hydrophobic organic pollutants, such as polycyclic aromatic hydrocarbons in the environment, generally have a large partition coefficient of octanol-water, low water solubility, and are easy to be distributed into environmental organic matter, which can be enriched in living organisms [1]. Aromatic compounds of benzene and its derivatives as an important chemical material have been widely used in industries, while they will endanger human health when the industrial wastewater discharges into the environment directly. Furthermore, many kinds of benzene series have been classified into the list of priority control pollutants in the world. Among them, nitrobenzene (NB) not only has strong stability but also has strong biological toxicity. It can affect humans by inhalation, dermal and oral routes of exposure [2]. It also can cause methemoglobinemia, fatigue, dizziness, headache, and nausea [3]. In the industrial wastewater streams, the concentration of NB has been detected as high as 35–140 mg/L [4,5]. The national institute for occupational safety and health recommends an occupational exposure limit of 23 mg/m^3^ of NB for a 10 h workday. In China, the NB emission limit for industrial wastewater and standard limit for surface water sources of centralized drinking water are 2.0 mg/L and 0.017 mg/L, respectively. So, it is urgent to treat the NB wastewater. Methods are available for treating NB wastewater mainly including photocatalytic degradation [6,7,8,9,10,11,12,13], ozonation biodegradation [14,15,16,17,18], and adsorption [19,20,21,22,23,24,25,26,27,28,29]. Adsorption is the most common treatment for removing NB from wastewater, and activated carbon is the dominating adsorbent material. In recent years, some materials possessing hydrophobicity have attracted great interest for fundamental research and potential application in water-oil separation.

Polystyrene is a major material type in microplastics, and it has a stronger sorption capacity for aromatic compounds due to π–π interactions [30,31]. The surface of polystyrene is hydrophobic and contains a relatively large number of active aromatic sites that allow π–π interactions between aromatic rings to take effect, which means that the aromatics pollutants, such as nitroaromatics and azo-dyes in water should be easily adsorbed by polystyrene carriers [32]. Juan Wang [33] found that polystyrene can accumulate phenanthrene, nitrobenzene, and naphthalene in water. Furthermore, polystyrene microplastics had a size effect on the adsorption of phenanthrene and nitrobenzene [34]. Moreover, the polystyrene microspheres loaded zero-valent iron had good potential application in the reductive degradation and emergency rescue of nitrobenzene contamination [32]. However, it is inconvenient to separate polystyrene from the emulsion after polymerization for its small size. Besides, polystyrene is superhydrophobic and difficult to disperse directly in water, which limits its application in water pollution treatment.

Cellulose is the most abundant and renewable natural polymer compound in nature. It is naturally degradable that is considered to be an environmentally friendly material [35,36]. Cellulose is a linear polymer formed by a large number of d-glucopyranose anhydrides (AGUs) bonded to each other by β-(1-4) glycosides [37]. Each AGU has a primary alcoholic hydroxyl group and two secondary alcoholic hydroxyl [38]. A large number of hydroxyl groups are advantageous for various grafting reactions, making cellulose-based adsorbent become a research hot in recent years. However, hydroxyl groups in the surface of cellulose are extremely hydrophilic which limits its mechanical strengthening effect. Moreover, cellulose tends to swell in the aqueous solution, leading to adsorbent damages and a low recovery rate. Besides, the weak affinity for organic pollutants caused by the non-polarity of cellulose makes it ineffective for the treatment of organic pollutants [39,40,41].

Consequently, we propose to synthesize a new adsorbent combining cellulose and styrene. On the one hand, cellulose as a matrix provides a lot of points for the attachment of polystyrene, so that the adsorbent has a certain macroscopic size for easy separation after preparation and recovery. On the other hand, cellulose can be used as a hydrophilic part to alleviate the superhydrophobicity of polystyrene and improve its dispersibility in water. The aim of this study was to prepare a kind of derivative cellulose adsorbent possessed of excellent hydrophobicity using styrene as the hydrophobic monomer. The free radical polymerization was applied to synthesize cellulose-styrene copolymer (cellulose-St). Cellulose- and alkali-treated cellulose (cellulose-OH) had no adsorption effect on NB, while cellulose-St showed potential for adsorption of NB, which was proved by adsorption experiments. Then, corresponding adsorption experiments were carried out to study its adsorption behaviors. It was noted that dynamic adsorption experiments can better reflect the potential of an adsorbent material for industrial applications, which were also applied to study the adsorption performance of cellulose-St. Furthermore, we demonstrated that cellulose-St can keep good adsorption properties for NB under such harsh environment as strong acids, strong alkalis, and organic solvents. Additionally, the experiments were also conducted regarding the repeated use of cellulose-St for adsorbing NB.

## 2. Materials and Methods

### 2.1. Materials

Cellulose with an average particle size of 90 µm was provided by Sinopharm Chemical Reagent Co., Ltd. (Shanghai, China), and dried under vacuum at 105 °C for 24 h prior to use. Styrene (≥99%) was dried over CaH_2_ and distilled under vacuum, then stored at 4 °C. All other solvents and chemicals were analytical reagent grade, supplied by Aladdin (Shanghai, China), and used without further purification. Ultrapure water (18.2 MΩ cm of resistivity) used in all experiments was purified by Hitech Smart-S Water Purification System (Shanghai, China) coupled with a reverse osmosis system.

### 2.2. Pretreatment

One gram cellulose powder was dispersed in 10 mL alkali solution (12.5 g dissolved in 1 L water). Then, the beaker containing cellulose and alkali solution was sealed with plastic wrap, which was exposed to ultrasonic below 30 °C for 1.5 h at a constant output power of 100 W. After that the mixture was centrifuged (5424R, Eppendorf, Hamburg, Germany) for 20 min at 10,000 rpm and 25 °C, it was washed till neutrality by water. The precipitate was frozen overnight at −78 °C in an ultra-low-temperature freezer (Zhongke Meiling Cryogenics Co., Ltd., Hefei, China) and then freeze-dried under 80 mbar pressure and −52 °C temperature (Beijing Boyikang Experimental Instrument Co., Ltd., Beijing, China). Freeze-drying is carried out to remove water traces for 24 h. The product was stored in the oven under vacuum (Shanghai Heqi Glassware Co., Ltd., Shanghai, China) at 25 °C to limit water uptake. The pretreated product named alkaline cellulose (cellulose-OH) was obtained.

### 2.3. Preparation of Cellulose-Styrene Copolymer

One gram of alkali cellulose was uniformly dispersed in 15 mL of ultrapure water on a three-necked flask, and the mixture was heated to 50 °C. After 30 min, 4 g styrene monomer followed by 0.1 g *N,N*′-methylenebis(acrylamide), 0.2 g K_2_S_2_O_8_ and 1 g Tween 80 as redox pair initiator were introduced with stirring vigorously for 1 h. Nitrogen atmosphere (10 mL min^−1^) was kept throughout the process. The polymerization reaction was performed under inert nitrogen gas atmosphere, stirring at 250 rpm and heating at 80 °C (Vacuum drier, Shanghai Heqi Glassware Co., Ltd., Shanghai, China). The reaction was completed after 5 h by decreasing the temperature to room temperature. The mixture was washed with water and ethanol in turn until the supernatant was clarified. The styrene–cellulose copolymer was covered by filter paper and washed with toluene in a Soxhlet extractor (Shanghai Aron Biochemical Instrument Co., Ltd., Shanghai, China) for 24 h successively in order to remove the homopolymer. Finally, the obtained cellulose–styrene copolymer (cellulose-St) was dried in a vacuum drier (Shanghai Heqi Glassware Co., Ltd., Shanghai, China) at 50 °C until a constant weight was reached.

### 2.4. Adsorption of Nitrobenzene (NB) on Cellulose-St

One milliliter methanol was added to stabilize the stock solution, which was subsequently diluted with deionized water to obtain the desired concentrations (5–35 mg L^−1^). The concentration of NB was determined by UV–Vis absorption spectrophotometry at 267 nm.

Fifty milliliters of aqueous NB solution with an initial concentration of 10 mg L^−1^ was added to a 100 mL conical flask equipped with a magnetic rotor (Shanghai ZhiCheng Analytical Instrument Co., Ltd., Shanghai, China), which was set at 250 rpm. Then, 10 mg of suction-dried cellulose-St was added to the flask at room temperature (25 °C). The concentration of NB in the solution at different times was measured (Shimadzu (China) Co., Ltd., Shanghai, China), and the adsorbed amount of NB was calculated based on the difference between the NB concentration in the original solution and the current solution.

The dynamic fixed bed studies were conducted to evaluate the effects of bed depths and flow rates on breakthrough curves in a laboratory-scale glass column with an internal diameter and length of 10 mm and 100 mm, respectively. 0.5 g of the modified cellulose–styrene - was fixed in column. Feeding solutions with initial NB concentrations (10 mg L^−1^, 30 mg L^−1,^ and 50 mg L^−1^) was pumped into the column by a peristaltic pump (Longer Precision Pump Co., Ltd., Hebei, China) at room temperature, with a flow rate of 10 mL min^−1^ at room temperature and pH = 7.0. Effluent samples were collected at defined time intervals and subsequent measure process was same as batch mode binding experiments.

The NB uptake ($Q_{e}$) is calculated as follows:(1)$$Q_{e}=\left(C_{0}-C_{e}\right)V/m$$
where $C_{0\;}$and $C_{e}$ (mg L^−1^) are the concentrations of NB before and after adsorption, respectively, V (L) is the volume of the NB solution, and m (g) is the weight of the adsorbent.

To test the performance of reuse of cellulose-St, the used cellulose-St was collected and washed with ethanol by vigorously stirring several times. Furthermore, the recycled cellulose-St was dried to repeated adsorption. Moreover, the adsorption capacity was determined as described above, expressed as a percentage of that without regeneration. The protocol was repeated ten times.

The recyclability rate ($R_{e}$) of cellulose-St is calculated as follows:(2)$$R_{e}=Q_{r}/Q_{e0}\times 100\%$$
where $Q_{r}$ and $Q_{e0}$ (mg L^−1^) is the adsorption capacity of reused and first used cellulose-St for NB, respectively.

### 2.5. Characterization

#### 2.5.1. Fourier Transform Infrared Spectroscopy (FTIR)

FTIR analysis was performed to determine the functional groups present in cellulose by using Nicolet iS50 (Thermo Fisher Scientific, Waltham, MA, USA) in the range from 500 cm^−1^ to 4000 cm^−1^. The samples were analyzed by attenuated total reflectance (ATR, Thermo Fisher Scientific, Waltham, MA, USA) in which the samples were placed on the evanescent wave on the ATR crystal, through which infrared beam gave the data to the detector.

#### 2.5.2. X-ray Diffraction (XRD)

XRD analysis was performed for neat cellulose, cellulose-OH, and cellulose-St. The samples were placed in a 2.5 mm deep cell, and measurements were performed with a MiniFlex 600 X-ray diffractometer (Rigaku, Tokyo, Japan) equipped with a detector. The operating conditions of the refractometer were: copper K_α_ radiation (*λ* = 1.5418 Å), 2θ (Bragg angle) between 5° and 80°, step size 0.067°, and counting time 10 min.

The empirical crystallization index *CrI* proposed by Segal [42] is calculated as follow forms:

For natural cellulose (cellulose I),
(3)$$CrI/\%=\left(I_{002}-I_{amorph}\right)/I_{002}\times 100\%$$
where $I_{002}$ represents the maximum diffraction intensity of the main crystallization peak 002 at 2θ = 22.6°, and $I_{amorph}$ represents the diffraction intensity at 2θ = 18°.

For regenerated cellulose (cellulose II),
(4)$$CrI/\%=\left(I_{101}-I_{amorph}\right)/I_{101}\times 100\%$$
where $I_{101}$ is the maximum diffraction intensity of the 101 peak (2θ = 20.8°), and *I_amorph_* is the diffraction intensity of the amorphous portion at 2θ = 16°.

#### 2.5.3. Thermo-Gravimetric Analysis (TGA)

The thermal degradations of samples were observed by TGA using a Simultaneous Thermal Analyzer (STA) 449C (Netzsch, Selb, Germany). Weight loss and derivative thermogravimetry (DTG) curves were recorded for a 20 mg sample at a heating rate of 10 °C/min in the temperature range of 30–1000 °C under inert gas atmosphere (40 mL/min).

#### 2.5.4. Scanning Electron Microscopy (SEM)

The SEM was operated at 15 kV to observe the microstructure of the surface in order to determine the roughness of particles after synthesis reaction and study the homogeneity of their dispersion. The samples were coated with a gold layer in order to prevent the charring of the sample due to the electron bombardment. The SEM images were captured using Quanta 250 (FEI, Hillsboro, OR, USA).

#### 2.5.5. Carbon-13 Solid Nuclear Magnetic Resonance (^13^C NMR) Spectroscopy Analysis

^13^C NMR spectrum was recorded on AVANCE III 500 NMR spectrometer (Bruker, Faellanden, Switzerland). All spectra were recorded by using a combination of cross-polarization, high power proton decoupling, and magic angle spinning (CP/MAS) methods.

#### 2.5.6. Brunauer–Emmett–Teller (BET)

The Brunauer–Emmett–Teller specific surface area (BET) was determined by N_2_ physisorption using a Micromeritics ASAP 2020 (Micromeritics, Shanghai, China) automated system [43]. More than 0.1 g of each powder sample was degassed at the temperature of 115 °C for 4 h prior to the analysis followed by N_2_ adsorption at −196 °C. BET analysis was carried out with a relative vapor pressure of 0.01–0.3. The average pore size of the samples was estimated from the nitrogen desorption isotherm according to the analysis of Barrett–Joyner–Halendar (BJH) model [44].

#### 2.5.7. Contact Angle Measurements

The hydrophilic/hydrophobic natures of samples were evaluated at room temperature by measuring the contact angle of a small water drop (ca. 5 μL) using an Attension Theta contact angle meter (Biolin Scientific, Gothenburg, Sweden). Smooth surfaces were obtained by compacting dried samples under a pressure of 10 MPa. Water contact angle values were determined from Attension Theta Software.

## 3. Results and Discussion

### 3.1. Characterization

Cellulose powders as raw materials were firstly treated by NaOH solution to obtain the precursor (cellulose-OH) which possessed lower crystallinity, more pleasing specific surface area, and better reactivity than raw materials. Ultrapure water as a solvent for polymerization instead of organic solvent had the advantages of safety, wide range of sources, low cost, and environmental protection. Then, the surface interaction force was enhanced by an emulsifier, and the reaction system proceeded under the action of a radical initiator. In the process, cellulose and styrene underwent a disordered graft polymerization activity, including the self-polymerization of styrene. The free polystyrene would be removed by Soxhlet extraction of toluene in a subsequent treatment. The desired dried product was obtained by vacuum drying. Scheme 1 showed the brief proposed synthesis of the graft copolymers.

Further study of the microscopic morphology of the surface was carried out by scanning electron microscopy analysis (SEM). It can be seen from Figure 1a–c that the original cellulose surface still exhibited the characteristics of smooth, non-porous, and wrinkle-free at high resolution, and it had a tendency to agglomerate. After alkali treatment, the size of cellulose particles became smaller, with a relatively uniform dispersion, shown in Figure 1d–f.

Moreover, the surface had irregular wrinkles and pores. In the view of macroscopic observation shown in Appendix A, the volume of the cellulose had a significant increase after being pretreated by alkali compared to the original cellulose at the same mass, as a result of the alkali treatment succeeding in swelling the cellulose fibers. The swelling allowed for more accessibility to the cellulose structure by increasing the surface area while reducing cellulose crystallinity and the degree of polymerization [45]. Alkali was thought to enhance hydrogenolysis conversion of cellulose to C2–C3 polyols [46]. In cellulose-St (Figure 1g–i), it can be found that the pores in the surface disappeared, and the surface was almost entirely covered with spherical irregular polystyrene. It meant that the grafting reaction occurred uniformly on the surface and styrene was grafted onto cellulose in a non-fixed structure. Furthermore, in Appendix A, it was found that the volume had a sharp reduction when cellulose-OH was converted to cellulose-St, which meant the increase of density. It may be caused by an increase in molecular weight. Moreover, cellulose-St had no significant change in color compared to cellulose and cellulose-OH. The BET analysis (Appendix A and Appendix A) showed the specific surface area raised nineteen times as against the original cellulose after alkaline treatment, indicating that alkali may cause some pore structure on the surface of cellulose, which can be found in SEM images. The specific surface area (as shown in Appendix A) decimated after polymerization since the surface-active sites of cellulose were occupied by styrene monomer. Moreover, the abnormal pore size distribution of cellulose and cellulose-St shown in Appendix A could be explained by the low surface area and the agglomeration. The impact of the surface modification of cellulose with styrene on hydrophobicity was determined by water contact angle measurements and the results were reported in Figure 1j–l. It showed that the hydrophobicity was significantly improved compared to that of neat cellulose. The extremely low values of water contact angle for original cellulose and cellulose-OH were obviously due to their hydrophilic nature resulting from hydroxyl groups on the surface of cellulose. However, the water contact angle of cellulose-St had exceeded 134°. Styrene exhibited hydrophobicity due to its special aromatic structure, and the hydroxyl groups on the surface of the cellulose were substituted or covered by polystyrene chains after grafting, causing the hydrophobicity of cellulose-St.

The chemical structures of the original cellulose, alkali-treated cellulose (cellulose-OH), and the cellulose grafted with styrene (cellulose-St) were confirmed by FTIR and ^13^C NMR. Three curves in Figure 2a represented the FTIR spectra of the original cellulose, cellulose-OH, and cellulose-St, respectively, according to the legend. Compared to original cellulose, the positions of the infrared characteristic adsorption peaks of cellulose-OH had no obvious change, but the intensity and shape of peaks were different. The intensity of absorption peak of the cellulose-OH at around 3333 cm^−1^ which corresponded to hydroxyl groups decreased and absorption band broadened simultaneously. It indicated that some of the hydrogen bonds in the intermolecular hydrogen bonds between cellulose molecules structures were weakened by the alkalinity of sodium hydroxide [47]. Some new peaks appeared in cellulose-St. The adsorptions at 3064 cm^−1^ and 3024 cm^−1^ were attributed to the stretching vibration of C–H in the benzene ring, the adsorptions at 1598 cm^−1^, 1543 cm^−1^, and 1488 cm^−1^ were attributed to the stretching vibration of C=C in benzene ring, and the adsorptions at 754 cm^−1^ and 698 cm^−1^ were the deformation vibration of C–H in the benzene ring. Additionally, the O–H stretching peak of cellulose still existed in cellulose-St, suggesting that the chemical grafting of styrene did not consume all of the hydroxyl groups on the surface of the cellulose. The hydrogen-bonding interactions formed by the remaining free hydroxyl groups could be enhanced in the copolymer [48]. Furthermore, ^13^C NMR was used to determine the chemical structure of cellulose-St. The signals at δ = 105.08, 74.74, 72.60, 88.28, and 65.20 ppm were assigned to the carbons on the glucose ring at C1, C2, C3, C4, C5, and C6, respectively, which were consistent with Yin’s research on cellulose [49]. These signals can be found both in cellulose (Figure 2c) and cellulose-St (Figure 2d). Furthermore, new signals at δ = 46.11, 40.16, and 145.89 ppm agreed with C7, C8, and C9 of styrene, δ = 127.86 ppm (C10) was corresponding to five identical Cs on benzene ring of styrene. On the basis of the above results, it was proved that styrene was successfully grafted on the surface of cellulose, and the molecular structure of cellulose-St is given in Figure 2d.

The crystalline structures of cellulose, cellulose-OH, and cellulose-St were tested with XRD analysis. The XRD spectra were shown in Figure 2b, at the same time, the intensity of diffraction peaks was obtained and used to calculate the crystallinity listed in Table 1. The crystal form of primary cellulose was cellulose I, where the diffraction peaks at 14.4°, 16.4°, 22.6°, and 34.6° corresponded to the crystal plane of (110), (101), (002), and (040), respectively. After the alkali treatment, new diffraction peaks were formed at 12.0°, 20.8°, 21.7°, which were the characteristic diffraction peaks of the cellulose II crystal form. Cellulose structure consisted of crystalline areas and amorphous areas, and it will remain in a state in which the crystal region and the amorphous region coexist throughout the transition [46]. The crystallinity of cellulose decreased from 84.53% to 55.69% after pretreatment. The crystalline areas were partly decrystallized due to the presence of alkali, making the crystal form to transfer and the crystallinity to decrease. Besides, ultrasonic accelerated the penetration of alkali into cellulose, which greatly shortened the pretreatment time. Ultrasound can also change the fiber morphology, supramolecular structure, and molecular weight and distribution of cellulose, which had been reported in some studies [50,51,52,53,54]. The number of diffraction peaks was reduced and the intensity of the diffraction peak at 2θ = 22.6° had a sharp decrease after grafting. It revealed that styrene had a negative influence on the crystallinity of cellulose though there was no formula for calculating the crystallinity of cellulose styrene copolymer. Styrene had changed the crystallinity of pure cellulose to amorphous state, which was strong proof for the implementation of the polymerization reaction.

The thermal decomposition behaviors of cellulose, cellulose-OH, and cellulose-St were examined by TGA and DTG, as shown in Appendix A. An initial weight loss was observed at lower temperatures in all of the three samples, which was ascribed to moisture vaporization. Moreover, cellulose-OH had the biggest loss in it for the more hydrophobic nature, while cellulose-St had the smaller. The cellulose and cellulose-OH had only similar decomposition peaks at around 340 °C when the decomposition temperature was above 120 °C. It also indicated that the alkali treatment did not change the thermal stability of the cellulose significantly. In contrast, cellulose-St exhibited two decomposition peaks at approximately 330 °C and 435 °C, which can be attributed to the thermal decomposition of the cellulose and styrene components of the copolymer. The thermal stability of cellulose-St was clearly improved compared to unmodified cellulose and cellulose-OH.

### 3.2. Adsorption

In order to study the practical application properties of the material, a series of adsorption tests were carried out. The adsorption time to reach an equilibrium of NB by the cellulose-St adsorbent was studied and shown in Figure 3a. Obviously, the removal rate of NB rose rapidly in the initial and increased gently until 5 h. Continuing to extend the contact time, the amount of adsorption no longer rose, indicating the adsorption equilibrium was reached at 5 h.

The intraparticle diffusion model is commonly used to analyze the control steps in the reaction, and to determine the internal diffusion rate constant of the adsorbent particles, by which the adsorption mechanism of NB on the cellulose-St adsorbent can be identified. The intraparticle diffusion equation is expressed as:(5)$$Q_{t}=k_{id}t^{1/2}+C$$
where $Q_{t}$ is the adsorption amount, mg/g; *k_id_* is the intraparticle diffusion rate constant, mg/(g·min^1/2^); C is the constant related to the thickness of the boundary layer, mg/g; *t* is the reaction time, min.

The fitting curves of the intraparticle diffusion model for the adsorption of NB by cellulose-St adsorbent are shown in Figure 3b, and the relevant parameters were listed in Table 2. It can be found that the adsorption process was a continuous segmentation with three stages. The first part ($Q_{t1}$) was related to surface diffusion, the second part ($Q_{t2}$) was internal diffusion process, and the third part ($Q_{t3}$) was the equilibrium of adsorption and desorption. At the initial stage of adsorption, the adsorption amount of NB was rapidly increased by internal diffusion of the liquid, then it was diffused into the interior of the adsorbent particles and adsorbed. The intraparticle diffusion was the rate control step of this process. With the progress of adsorption, the adsorption active sites were gradually reduced, the concentration of NB was gradually decreased, and the diffusion rate began to slow down. At last, the solid–liquid phase distribution turn balanced, the adsorption saturation was achieved, and the adsorption amount was no longer increased. The C value of the three fitting straight lines was not equal to zero, indicating that the internal diffusion was not the only step to control the adsorption. In other words, two or more adsorption rate-limiting steps were likely to take place in the entire adsorption process.

The adsorption kinetics can be analyzed using pseudo-first-order and pseudo-second-order kinetic models as follows:

Pseudo-first-order:(6)$$\ln\left(Q_{e}-Q_{t}\right)=\ln Q_{e}-k_{1}t$$

Pseudo-second-order:(7)$$t/Q_{t}=t/Q_{e}+1/\left(k_{2}Q_{e}^{2}\right)$$
where $Q_{e}$ (mg/g) and $Q_{t}$ (mg/g) are the amounts of NB sorption at equilibrium and at the contact time t (min); $k_{1}$ (min^−1^) and $k_{2}$ (g·mg^−1^·min^−1^) are the corresponding kinetic rate constants.

The adsorption kinetics models were plotted in Figure 3c,d. The fitting results of the two different models were summarized in Table 3. According to the correlation coefficient (R^2^), the adsorption of NB of cellulose-St adsorbent obeyed both the pseudo-first-order and pseudo-second-order kinetic model. The correlation coefficient of pseudo-second-order kinetic model (R^2^ = 0.9975) was closer to 1 than pseudo-first-order model (R^2^ = 0.9843). Therefore, pseudo-second-order kinetic model can reflect the mechanism of the adsorption process more accurately.

The adsorption isotherms data of NB over cellulose-St adsorbent at various temperatures were simulated by Langmuir and Freundlich models. As illustrated in Appendix A and Appendix A), the high R^2^ values demonstrated that NB adsorption on cellulose-St both obeyed the Langmuir and Freundlich models. In addition, the NB adsorption capacity decreased along the temperature, suggesting the adsorption was an endothermic process. Based on the Langmuir model, the high adsorption capacity of NB for cellulose-St was 131.06 mg/g.

Dynamic adsorption was more capable of reflecting the potential of engineering applications of material than static adsorption. Hence, the practical availability of cellulose-St on the adsorption of NB was not only studied with static experiments but also dynamic adsorption, and the results were shown in Figure 4. Dynamic adsorption tests were performed with three different initial concentrations (10 mg/L, 30 mg/L, and 50 mg/L) at atmosphere, and filler had an aspect ratio of 1:8. The treatment amount of NB with a feed concentration of 10 mg/L was 1.275 L/g. When the feed concentration was increased to 30 mg/L and 50 mg/L, the treatment amount decreased to 0.75 L/g and 0.5 L/g. Before reached the breakthrough point, no NB was detected in the effluent at the initial concentration of 10 mg/L, while it was not when the initial concentration increased to 30 mg/L and 50 mg/L. Furthermore, the higher the initial concentration of NB, the shorter time to reach penetration point and final balance. Apparently, a high inlet velocity of wastewater resulted in shorter contact time for NB and cellulose-St and small average adsorption sites, which may be reflected by the increased export concentration with the increased input initial concentration before reaching individual throughput. The lower the initial concentration of the feed liquid in a certain range, the larger the treatment amount, but the initial concentration of the feed liquid was not unlimited. The mass transfer driving force will be too small when the initial concentration of the feed liquid was too low, resulting in a decrease in adsorption capacity and effective utilization rate. All in all, cellulose-St can be applied to the wastewaters with a low content of NB.

In practical applications, it was often not a simple aqueous solution, but also various harsh conditions, so the environmental adaptability of adsorbent was essential. To examine its environmental adaptability, cellulose-St was placed in a variety of harsh environments. Cellulose-St was immersed in strong acid (pH = 2, pH = 4), strong alkali (pH = 10, pH = 12), and organic solvent (n-octane, n-heptane) environment for 24 h, respectively. Then, the water contact angles and adsorption amount for NB of processed cellulose-St were measured. From Figure 5, we can see that the samples after the above several treatments still exhibited excellent hydrophobic properties. The treated sample was applied to the adsorption of NB, and the change in the adsorption value was within 1 mg/g. Therefore, cellulose has excellent environmental adaptability and can be applied to adsorb NB in various harsh water environments.

Through the above experiments, we found that cellulose-St showed an ability for adsorbing NB. However, a blank control experiment showed that cellulose and cellulose-OH had no adsorption effect on NB (Appendix A). Therefore, it can be concluded that the introduction of styrene was the key to the occurrence of adsorption. In order to explain the specific effect of styrene on adsorption and reveal the adsorption mechanism, several samples with different water contact angles were successfully prepared by adjusting the amount of styrene added during the preparation process, and the adsorption was studied under the same conditions. The water contact angles of the samples and the corresponding adsorption amount were shown in Figure 6. The water contact angles ranged from 88.44° to 146.61° with the different grafting rates. Within a predetermined range, the higher the hydrophobicity, the larger the corresponding adsorption amount, but when the water contact angle reached the threshold, the adsorption amount will decrease. In a suitable range of hydrophobicity, nitrobenzene exhibits repulsion due to its opposite polarity to water, forcing nitrobenzene to move closer to the adsorbent and undergo physical and chemical adsorption. While after the hydrophobicity reached a certain level, the contact area between the aqueous solution and the adsorbent was reduced, which affected the contact of the nitrobenzene in the aqueous solution with the adsorbent, thereby a decline of the adsorption amount was observed.

Based on the above results, we speculate that the mechanism for cellulose-St to adsorb NB is shown in Scheme 2. In the adsorption process of NB onto cellulose, the NB and cellulose-St are compelled to attach firstly by hydrophobic interaction, due to the difference in polarity between the two and water molecules. Nitrobenzene is π-electron-deficient with a strong dipole moment. Therefore, it acts as a π-acceptor with cellulose-St surface sites. Nitrobenzene oxygens are H-bond acceptors with the remained hydroxyl groups, similar to nitroaromatic adsorptions on graphene oxide [55]. The rich aromatic regions on the cellulose-St supported by styrene chain are π-electron donors for NB stacking. In summary, NB adsorption over a wide pH range by multiple mechanisms. π–π interactions may predominate because NB possesses polarized aromatic π-electron systems interactive with the extended aromatic regions of cellulose-St. H-bonding interactions between surface functional groups and –NO_2_ groups can contribute to it.

### 3.3. Reusability

The reusability of the adsorbent was very essential to employ in the environmental region. The used cellulose-St was separated from the aqueous solution by filtration and washed with ethanol several times. After being dried, the used cellulose-St was applied for the recycle test of NB adsorption, and the results were shown in Figure 7. It could be found that cellulose-St still had an adsorption capacity of 90.1% of the original adsorption capacity even after ten cycles. The results demonstrated that cellulose-St had excellent recyclability and could be regarded as a potential adsorbent for removing NB from aqueous solutions. Furthermore, the reusability of cellulose-St was compared with other adsorbents reported previously as shown in Table 4. Due to its lower environmental impact, ethanol was an effective method for desorption of NB compared to other organic solvents such as methanol, toluene, and acetone. The recyclability rate obtained from this work were found to be much higher than that of some adsorbents in Table 4.

## 4. Conclusions

Furthermore, Appendix A listed the adsorption performance, pH adaption, and reusability of different activated carbons, while their adsorption performance on nitrobenzene varied significantly. As shown in Appendix A, the commercial granular activated carbon (87 mg/g), acidic oxygen functionalized activated carbon (42 mg/g), and magnetically separable porous carbon microspheres (97 mg/g) displayed a lower adsorption capacity on nitrobenzene than our materials (131.06 mg/g). Although the activated carbon from rice husk and activated carbons prepared from vegetable waste exhibited higher adsorption capacity, they showed lower pH adaption and reusability than our materials. In addition, as a green material, cellulose-based materials are sustainable and environment-friendly. Therefore, cellulose-St obtained in this work could be considered as an effective potential adsorbent for the removal of NB from aqueous solutions in the future.

Cellulose-styrene copolymers (cellulose-St) were successfully prepared by emulsion polymerization which was cost-effective, easy-to-operate, and considered to be a green approach. Due to the presence of benzene rings, cellulose changed from hydrophilic to hydrophobic, the water contact angles ranged from 98° to 145° with the different grafting rates. Moreover, it provided cellulose with the ability to adsorb nitrobenzene (NB). The adsorption kinetics fitted well to the intraparticle diffusion model and pseudo-second-order (R^2^ > 0.99) kinetic model. Furthermore, dynamic adsorption test showed that cellulose-St had the continuous separation potential for NB in an aqueous solution. The breakthrough points for the initial NB concentration of 10 mg/L, 30 mg/L, and 50 mg/L were 1.275 L/g, 0.75 L/g, and 0.5 L/g. The adsorption was mainly achieved by π–π stacking interaction and hydrophobic interaction. Moreover, cellulose-St exhibited environmental adaptability. Cellulose-St could maintain good hydrophobicity and stable adsorption capacity for NB even in the presence of strong acids, strong alkali, or high concentrations of organic solvents. Cellulose-St also showed excellent recyclability that could keep up to 90.1% of original adsorption capacity after ten cycles. Therefore, cellulose-St obtained in this work had the potential to adsorb NB in the future.

## Data Availability

The data presented in this study are available on request from the corresponding author.

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
