# Peer review of "Preparation of a Novel Cellulose–Styrene Copolymer Adsorbent and Its Adsorption of Nitrobenzene from Aqueous Solutions"

_polymers, 2021, doi:10.3390/polym13040609_

Round 1
Reviewer 1 Report
The article titled "Preparation of a novel cellulose-styrene copolymer adsorbent and its adsorption of nitrobenzene from aqueous solutions" presents the development of a new absorbent for the removal of NB from wastewater. This is an interesting article where it would be good to say more in theory about NB concentrations in wastewater and what individual concentrations cause and at what NB concentration there is no danger to human health. It also lacks whether we can use the adsorbent cellulose styrene copolymer to remove a sufficient amount NB in certain wastewaters.
The article lacks economic justification for the use of cellulose - St and applicability compared to a commonly used absorbent such as activated carbon.
In section 3.2, I suggest that the same values as in Figure 3 be used in the interpretation of Figure 3.
Author Response
1) The article lacks economic justification for the use of cellulose - St and applicability compared to a commonly used absorbent such as activated carbon.
Response: Thank you for your useful comments for improving our manuscript. Table R1 listed the adsorption performance, pH adaption and reusability of different activated carbons, while their adsorption performance on nitrobenzene varied significantly. As shown in Table R1, the commercial granular activated carbon (87 mg/g), acidic oxygen functionalized activated carbon (42 mg/g) and magnetically separable porous carbon microspheres (97 mg/g) displayed a lower adsorption capacity on nitrobenzene than our materials (131.06 mg/g). Although the activated carbon from rice husk and activated carbons prepared from vegetable waste exhibited higher adsorption capacity, they showed lower pH adaption and reusability than our materials. In addition, as a green material, cellulose-based materials are sustainable and environment-friendly. Thus, it can be concluded that cellulose-styrene copolymer could be promising an alternatives for the remediation of nitrobenzene. The related discussion was added in the revised manuscript (Section 3.3).
Table R1 Nitrobenzene adsorption performance over reported activated carbon materials
Absorbents |
pH adaption |
Adsorption capacity (mg/g) |
Reusability |
References |
Commercial granular activated carbon |
- |
87 |
- |
[1] |
Activated carbon from rice husk |
4-10 |
446 |
- |
[2] |
Acidic oxygen functionalized activated carbon |
- |
42 |
- |
[3] |
Activated carbon from wood |
- |
238 |
- |
[4] |
Activated carbons prepared from vegetable waste |
2-12 |
476 |
58.7% after 4 runs |
[5] |
Activated carbons prepared from vegetable waste |
2-12 |
490 |
62.8% after 4 runs |
[5] |
Magnetically separable porous carbon microspheres |
- |
97 |
84% after 5 runs |
[6] |
cellulose-styrene copolymer |
2-12 |
131 |
90.1% after 10 runs |
This work |
2) In section 3.2, I suggest that the same values as in Figure 3 be used in the interpretation of Figure 3.
Response: We are sorry to make the mistakes in our text and we have revised in the manuscript (Section 3.2)

Reviewer 2 Report
In this paper by Yang and colleagues, the authors studied how the modification of cellulose changes its properties. The results are in line with the journal Polymers. However, there are issues, which must be tackled before the paper can be accepted for publication. Please find the suggestions below:
1) "Water used in all experiments was purified by HHitech Smart-S (Shanghai, China)" (Lines 94-95). The quality of water may have a large influence on the obtained results. What was the process used by the machine to purify it? What was the electrical conductivity of water upon purification?
2) The experimental section lacks many details necessary for the reproduction of the study. Since then the findings cannot be verified and other scientists cannot build on them, inevitably the article will be of a limited impact if not corrected. For instance:
- "Sodium hydroxide solution with a mass fraction of 12.5% was prepared for use at first. And an appropriate amount of cellulose powder was dispersed in an alkali solution at a mass ratio of 1:10." (Lines 97-99). Please specify the amounts.
- "Then the beaker containing cellulose and alkali solution was sealed with plastic wrap, which was exposed to ultrasonic at a certain temperature for 1.5 h in the next step." (Lines 99-100). What were the sonication parameters? Such processing produces high power density cavitation, which may deteriorate the material in time, so it is essential to provide the experimental conditions. Besides that, the use of a certain types of sonicators results in rapid heating of the homogenized mixture. This brings a question of how the temperature of the mixture was monitored.
- "After that the mixture was centrifuged" (Line 101) - no centrifugation parameters, which determine the amounts and chemical composition of pellet and supernatant
- SEM acceleration voltage not reported
- etc.
There are many more examples of such shortcomings, but, due for the sake of brevity, I invite the authors to find and correct them themself.
3) "Moreover, cellulose-St exhibited excellent environmental adaptability that it could maintain its hydrophobicity and adsorption ability for NB in strong acids, strong alkalis, or organic solvents" (Lines 19-21). Are the authors claiming that cellulose does not hydrolyze in acidic conditions?
4) Miller indices should be added to XRD patterns in Fig. 2 (patterns, not "spectra").
5) A serious concern in this study is that error analysis was neglected. There are no error bars, which puts in doubt if these results are statistically significant. Please conduct more experiments to obtain the statistics and verify the claims of this paper.
Author Response
1) "Water used in all experiments was purified by HHitech Smart-S (Shanghai, China)" (Lines 94-95). The quality of water may have a large influence on the obtained results. What was the process used by the machine to purify it? What was the electrical conductivity of water upon purification?
Response: We are sorry to not describe the detailed information on the water used in our experiments. The ultrapure water used in our work was purified by HHitech Smart-S Water Purification System (Shanghai, China) coupled with a reverse osmosis system and the resistivity of the water was about 18.2 M Ω .cm. We have revised it in the revised manuscript.
2) The experimental section lacks many details necessary for the reproduction of the study. Since then the findings cannot be verified and other scientists cannot build on them, inevitably the article will be of a limited impact if not corrected. For instance:
- "Sodium hydroxide solution with a mass fraction of 12.5% was prepared for use at first. And an appropriate amount of cellulose powder was dispersed in an alkali solution at a mass ratio of 1:10." (Lines 97-99). Please specify the amounts.
- "Then the beaker containing cellulose and alkali solution was sealed with plastic wrap, which was exposed to ultrasonic at a certain temperature for 1.5 h in the next step." (Lines 99-100). What were the sonication parameters? Such processing produces high power density cavitation, which may deteriorate the material in time, so it is essential to provide the experimental conditions. Besides that, the use of a certain types of sonicators results in rapid heating of the homogenized mixture. This brings a question of how the temperature of the mixture was monitored.
- "After that the mixture was centrifuged" (Line 101) - no centrifugation parameters, which determine the amounts and chemical composition of pellet and supernatant
- SEM acceleration voltage not reported
- etc.
There are many more examples of such shortcomings, but, due for the sake of brevity, I invite the authors to find and correct them themself.
Response: We are sorry to make these mistakes in our manuscript and we have added more details on the preparation and characterization of the material, as well as the description of the adsorption process so that our work can be reproduced by others, which can be seen in Section 2.2-2.4.
3) "Moreover, cellulose-St exhibited excellent environmental adaptability that it could maintain its hydrophobicity and adsorption ability for NB in strong acids, strong alkalis, or organic solvents" (Lines 19-21). Are the authors claiming that cellulose does not hydrolyze in acidic conditions?
Response: Based on the previous studies [7-11] [1-5], the cellulose can be depolymerized under strong acids through hydrolyzation. But the hydrolyzation only happens at rather high level of acids. For example, the cellulose can be hydrolyzed in 64 wt% sulfuric acid (10.1 mol/L).Notably, in our experiments, even in the solution with pH of 2, the H+ concentration was only 0.01 mol/L, which was one XX of that in the acid hydrolyzation system. In addition, according to our results, under the acid condition (pH=2), no evident decline on the NB adsorption performance over the materials was observed compared to those under other conditions. Thus, we assuredly supposed that cellulose-St exhibited excellent stability and environmental adaptability.
4) Miller indices should be added to XRD patterns in Fig. 2 (patterns, not "spectra").
Response: We are sorry to make the mistakes and we have added the Miller indices in the XRD patterns in the revised manuscript.
5) A serious concern in this study is that error analysis was neglected. There are no error bars, which puts in doubt if these results are statistically significant. Please conduct more experiments to obtain the statistics and verify the claims of this paper.
Response: In our work, except the dynamic adsorption, all the experiments were operated thrice and we have added the error bar on the data in the revised manuscript.

Round 2
Reviewer 2 Report
Thank you. The article may be accepted.
One minor mistake. In Fig. 6 the right Y-axis should read "Water contact angle". Please correct it at the proof stage.
This manuscript is a resubmission of an earlier submission. The following is a list of the peer review reports and author responses from that submission.